# The external realities of people with type 2 diabetes—Understanding disease perspective and self-management behaviour via Grounded Theory Approach

Yogarabindranath Swarna Nantha[1,2,3]*, Azriel Abisheg Paul Chelliah[2,3], Shamsul Haque[4], Gan Kim Yen[2], Anuar Zaini Md. Zain[1]

1 Clinical School Johor Bahru, Jeffrey Cheah School of Medicine and Health Sciences, Monash University Malaysia, Subang Jaya, Malaysia, 2 Non-Communicable Disease Department, Seremban Primary Health Care Clinic, Seremban, Malaysia, 3 Sibu Hospital, Sibu, Malaysia, 4 Department of Psychology, Jeffrey Cheah School of Medicine and Health Sciences, Monash University Malaysia, Subang Jaya, Malaysia

* yoga.rabindranath@monash.edu

**Data Availability Statement:** All relevant data are within the manuscript and its Supporting information files.

## Abstract

### Background

Qualitative strategies can uncover the relationship between the external realities of people living with type 2 diabetes (T2D) and the barriers that are associated with disease self-management. Information from in-depth interviews (IDI) and focus group discussions (FGD) can be used to devise psychological models that could potentially facilitate behaviour changes in people with T2D. We aim to identify salient factors that govern the external realities of people with T2D in relation to disease management.

### Methods

A qualitative study was conducted at a regional primary care clinic in Malaysia using a Grounded Theory Approach. People with T2D were recruited through purposeful sampling to determine their living experiences with the disease. A total of 34 IDIs with 24 people with T2D and 10 health care professionals, followed by two FGDs with people with T2D, were conducted.

### Results

Three major processes that arbitrate self-management practices include– 1) external reality, 2) internal reality, 3) mediators of behaviour. Within the context of external reality, three important sub-themes were identified—intrinsic background status, personal experience, and worldview. Lifestyle habits of persons with T2D play a central role in their disease management. Another common recurring concern is the issue of a low-quality food environment in the country. More importantly, individuals with T2D have a high degree of expectations for a more person-centered approach to their illness.

**Funding:** YSN received funding from the MOH-NIH grant (Grant No.:91000440) from the Ministry of Health Malaysia. The funders had no role in study design, data collection and analysis, decision to publish, or preparation of the manuscript.

**Competing interests:** The authors have declared that no competing interests exist.

## Conclusions

We identified modifiable and non-modifiable behavioural factors that influence the daily living environment of people with T2D. This information can be used to customize the management of T2D through targeted behavioural interventions.

## Introduction

Malaysia has witnessed an 80% increase in the prevalence of diabetes over the span of just a decade [1]. These numbers continue to grow as type 2 diabetes (T2D) trends saw a sharp increase from 11.6% in 2011 to 18.3% in 2019 [2, 3]. Contrary to expected outcomes, widely accessible services at public primary care clinics did not appear to translate into proportionate improvements in glycaemic control in this population [4]. This situation is evidenced by the stagnation of blood glucose control in people with T2D between 2009 and 2012 at an average HbA1c level of 8.2%) [5]. The reasons for this predicament are scarcely explained in literature. However. experts believe the principal reason for these patterns can be traced back to lapses in medication taking, food choices, and physical activity [4]. Therefore, a focused qualitative inquiry within this context might help uncover specific pitfalls in disease self-management.

Good self-management practices (engagement in treatment plan, dietary measures, and exercise) can increase glycaemic stability in people with T2D [6]. However, T2D self-management behaviour relies heavily on the habits that permeate the daily life of an individual with T2D. Consequently, self-management practices require changes to be made to the belief system that characterizes their 'living environment'. This obstacle remains the single most crucial challenge in T2D disease management [7–9]. Hence, there is a need for a qualitative exploration of the practicality of broadly accepted self-management practices routinely recommended for individuals with T2D.

This study builds upon a larger initiative to create a comprehensive qualitative framework that describes the characteristics of disease management in people with T2D [10, 11]. In previous papers, we have expounded on the drivers of 'internal realities' related to self-management behaviour [10, 11]. This analysis led to the discovery of 2 main categories– 1) internal reality (the inner psychological environment and personal volition), and 2) salient mediators that govern this behaviour (Fig 1). In the current article, we turn our attention to the exploration of the external realities (i.e., immediate physical and socio-cultural environment) of individuals with T2D and their influences on self-management behaviour.

## Material and methods

### Design

A qualitative Grounded Theory approach was chosen to identify behavioural patterns relevant to this study's objectives. An inductive line of qualitative inquiry was utilized to 1) assist the discovery of 'real-world' or 'naturalistic' themes, 2) ensure that codes remain closely grounded to emergent data, and 3) *remain open* to newer insight instead of forcing data to fit antecedent behavioural concepts seen in literature [12, 13]. This study protocol received approval from the Medical Research and Ethics Committee, Ministry of Health Malaysia (NMRR-18-151-39886) on the 30th of May 2018 and Monash University Human Research Ethics Committee on 18th of October 2018 (Project ID: 17062). All subjects in the study gave informed consent prior to participation.

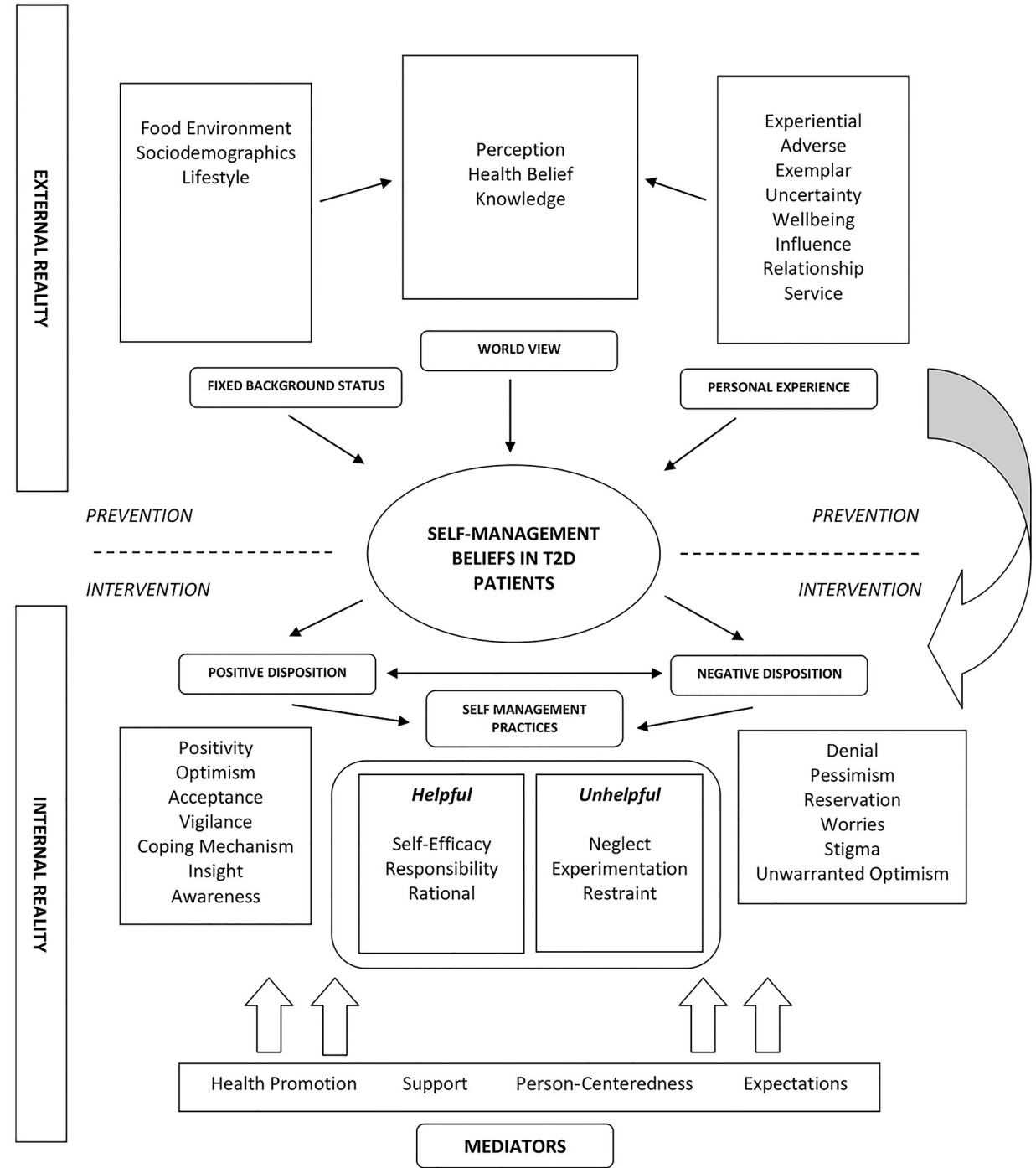

**Fig 1. Conceptual framework describing the external reality, internal reality, and mediators related to self-management of T2D.**

## Participants and setting

A purposeful sampling technique was utilized to recruit participants from the non-communicable disease (NCD) department at the Seremban Primary Care center in Malaysia, a large-scale general practice clinic within the district of Seremban. A maximum sampling strategy

was employed to achieve a good spread in terms of socioeconomic and disease control indicators (Tables 1 and 2). This step involves 1) selecting cases that accurately represent population diversity and 2) eliciting common behavioural patterns related to the dimension of interest [14]. A published study protocol provides a more detailed description of the research methods employed here [15].

## Data collection

All 34 in-depth (24 people with T2D and 10 health care professionals) interviews and 2 focus group discussions (comprising 8 people with T2D each session) took place between May 2018 to April 2019. Participants were interviewed in a designated research room within the premises of the clinic. A previously published protocol explains the interviewing technique and the topic guide (S1 File and S1 Fig) employed in this study [15]. To address the limitations of positionality, we minimized the framing of questions based on the pre-conceived notion of the interviewer by 1) taking up the role of a neutral observer, 2) remaining as far as possible an outside expert during interviews, and 3) allowing participants to freely express issues that were perceived relevant to them [16]. No new themes or categories were obtained after the 10th round of interviews. At this point, theoretical saturation was considered to have been achieved. However, establishing the validity of these categories required additional interviews with 14 patients where these narratives were further explored, enriched, and consolidated. The interviews were audiotaped and transcribed verbatim. One researcher (YSN) conducted all interviews.

## Data management and analysis

Once data collection was complete, individual transcripts were 1) proofread independently, 2) re-examined by the investigator, and 3) cross-compared using iterative methods. Proofread transcripts were then entered into a qualitative software (Atlas.TI qualitative analysis program, Version 7, Cincorn Systems Inc, 2008) and coded manually to identify specific themes by YSN. The data was coded by closely adhering to classical and constructivist Grounded Theory techniques [12, 17].

The coding of data involved a multi-stage process. First, all the codes seen in the transcripts were contrasted with each other to identify emerging themes. The first author (YSN) then analyzed the participants' responses from the qualitative study (YSN). This step was followed by a scheduled discussion with another author (SH) to categorize the data according to relevant emerging themes. This constant comparison strategy was carried out between researchers throughout the entire process of the study in an effort to refine the codes from the the transcripts. Transcripts were finally coded independently by two coders, and a list of emerging themes and categories was created. Excellent inter-coder consistency was found (Cohen's $\kappa$ = 0.84). Addtionally, the themes derived from this study were assessed by a panel of experts to establish coding uniformity and was found to have excellent content consistency (Cohen's $\kappa$ = 0.88).

## Trustworthiness

The trustworthiness of this study (S1 Fig; S1 Table) was assured by using different techniques of data collection through both in-depth interviews (IDI) and FGDs (methodological triangulation) [18]. Two researchers performed data analysis and interpretation (investigator triangulation). We chose to validate the information (data triangulation and negative case analysis) by comparing interviews that were obtained from diverse sources (people living with T2D and healthcare providers). Moreover, the authors interpreted, conceptualized, and revised behavioural constructs at various stages of the study via an iterative process (persistent observation).

**Table 1. Demographic details of T2D patients of the study.**

| Subject Characteristics | N |
|---|---|
| **Gender** | |
| Male | 15 |
| Female | 9 |
| **Age In Years** | |
| < 30 | 1 |
| 31–50 | 6 |
| 51–60 | 7 |
| 61–70 | 7 |
| > 70 | 3 |
| **Ethnicity** | |
| Malay | 12 |
| Chinese | 1 |
| Indian | 10 |
| Others | 1 |
| **Marital Status** | |
| Unmarried | 3 |
| Married | 19 |
| Divorced | 1 |
| Widow/Widower | 1 |
| **Work Status** | |
| Not working | 12 |
| Working Full-time | 8 |
| Working Part-time | 1 |
| Self-employed | 3 |
| **Level Of Education** | |
| Certificate level | 19 |
| Diploma | 3 |
| Bachelor's degree | 2 |
| **Diabetes duration (years)** | |
| < 5 | 5 |
| 6–10 | 3 |
| 11–15 | 4 |
| 16–20 | 2 |
| > 21 | 3 |
| **Number of medications** | |
| 3–5 | 12 |
| > 6 | 12 |
| **Type of medication** | |
| Oral only | 5 |
| Insulin only | 4 |
| Combination | 15 |
| **Glycaemic status** | |
| 6.5–8.0% | 3 |
| 8.1–9.0% | 9 |
| 9.1–10% | 4 |
| > 10% | 8 |

(*Continued*)

**Table 1.** (Continued)

| Subject Characteristics | N |
|---|---|
| **Complications** | |
| No | 17 |
| Yes | 7 |
| **Number of comorbidities** | |
| 1 | 13 |
| 2 | 4 |
| > 3 | 8 |

**Table 2. Demographic details of healthcare professionals involved in the study.**

| Subject Characteristics | N |
|---|---|
| **Profession** | |
| Diabetes educator (nurse) | 5 |
| GP | 3 |
| Pharmacist | 2 |
| **Mean Age In Years** | 34 |
| **Mean Years In Service** | 10 |
| **Ethnicity** | |
| Malay | 7 |
| Chinese | 1 |
| Indian | 2 |
| **Marital Status** | |
| Unmarried | 2 |
| Married | 7 |
| Divorced | 1 |
| **Level Of Education** | |
| Diploma | 3 |
| Bachelor's degree | 5 |
| Master's degree | 2 |

## Results

The meticulous process of systematically sifting through large amounts of data led to the development of multiple layers that define the distinct concept of external reality (Table 3, S2 and S3 Tables). Furthermore, a stepwise agglomeration of codes into subcategories and the subsequent deductive creation of unique themes appear to add more depth to the external factors that impact self-management behaviour in individuals with T2D. Based on the similarity of context, we also performed an iterative examination of narratives from FGDs (patients) and IDIs (healthcare providers). This process led to a more nuanced explication of each subcategory in keeping with dominant trends (vis- à-vis the experience of medication utilization and patient-doctor relationship) linked to self-management practices in literature [19, 20].

Fig 1 describes a theoretical framework for understanding how individuals with T2D determine overall self-management behaviour. The interaction between the external reality (the physical environment) and internal reality (the inner psychological environment) and mediators of behaviour can be best described as a dynamic cycle that continues to influence self-management behaviour in people with T2D (Fig 1). Notably, the results from this study are

outlined in the upper half of Fig 1, which illustrates a diagrammatic representation of elements closely associated with the external realities of persons with T2D.

Within the narrative of external reality, three critical themes were identified, namely 1) intrinsic background status, 2) personal experience, and 3) world view (Table 3). We developed the concept of 'intrinsic background status' by merging codes that reflect societal, cultural, and communal factors that shape the daily living experience of people with T2D. On the other hand, the theme classified as 'personal experience' was designed by taking into account various dimensions that portray the physical and emotional experiences relevant to living with T2D. Lastly, we define 'world view' as the personal perception of people with T2D about

**Table 3. Categories, definitions, frequency, and codes describing the external reality of T2D patients.**

| Category | Subcategory | Frequency | Codes |
|---|---|---|---|
| **Intrinsic background status** | Lifestyle | 212 | Medication taking behaviour |
| | | | Timing of medications |
| Social, cultural and community environment that shape the daily living experience of people with T2D | | | Practicality of dietary and diet advice |
| | Sociodemographics | 135 | The influence of age |
| | | | The influence of education |
| | | 44 | Cultural paradigmAffordability |
| | Food Environment | | Needing to eating out |
| | | | Poor food quality |
| | | | Solutions to existing milieu |
| **Personal experience** | Experiential | 196 | Experiencing physical symptoms |
| Physical and emotional experience pertinent to the phenomenon of living with diabetes | | | Response of body to medications |
| | Adverse | 156 | Experiencing side effects |
| | | | Coping with side effects |
| | Exemplar | 154 | Witnessing complications |
| | | | Emulating exemplary patients |
| | | | Becoming role models |
| | Uncertainty | 137 | Nature of disease |
| | | | Unpredictability of disease despite compliance |
| | Wellbeing | 90 | Positive effects of using modern medications |
| | | | Positive effects of using alternative medications |
| | Influence | 74 | Influence of family members |
| | | | Influence of friends |
| | Relationship with GPs | 49 | Positive disposition of doctors |
| | | | Negative disposition of doctors |
| | Service | 44 | Opinion about services |
| | | | Opinion about doctors |
| **World view** | Health Belief | 145 | Beliefs about complementary alternative medications |
| Self and society's point of view about disease | | | Beliefs about medication, illness, diet and exercise |
| | Knowledge | 125 | Mechanism for T2D |
| | | | Recognizing complications |
| | Perception | 74 | Perception of self |
| | | | Perception of others and society |
| | | | Devolving autonomy doctors |

themselves and their perceived opinion about what society thinks about T2D. We will discuss the details of these subcategories in the subsequent sections of this article.

## Intrinsic background status

**Lifestyle.** The collective characteristics of the living environment of individuals with T2D (within the backdrop of self-care activities) were classified under the subcategory called 'lifestyle'. These lifestyle features mainly influence medication compliance, followed closely by reservations over dietary and exercise measures.

In our study, persons with T2D miss taking their medication when they are out of their homes. They fail to bring along their medications and cite forgetfulness as the main reason for their shortcomings. Similarly, people leading hectic daily routines (being in a rush or immersed in completing daily chores) also tend to take their medications inconsistently.

> "I am a police officer. In a day, there are just far too many activities I am involved in. At one point, I am at the office, and before you know it, I am out doing field work. So sometimes I miss taking my medications because of that"
>
> (IDM 023)

Most individuals with T2D (corroborated by information in interviews with pharmacists competent in providing T2D self-management services) concur that ingrained lifestyle habits (mostly work-related) contribute to the difficulty of adhering to proper medication timing. They feel that taking oral medications is much more convenient than insulin, especially when they are on the go.

> "Maybe insulin yes, I do agree because we need to store it somewhere and excuse ourselves in order to find a suitable place to inject it. But tablets, they are not a problem at all. When you are sitting at a restaurant, while eating, you readily have it in your pocket. Right after having lunch, you just pop it in your mouth and drink a glass of water. That's about it"
>
> (IDM 011)

Persons with T2D also find it hard to engage in therapeutic lifestyle changes such as dietary measures and exercise. They feel that the recommended dietary advice is often impractical. Most importantly, they blame work constraints as a significant reason for restrictions on the type of food choices they can access. On a separate matter, maintaining exercise becomes a difficult habit to sustain as they are overwhelmed with work responsibilities and household chores.

> "Sometimes when you come home from work, you don't really feel like doing exercise, especially when you come home late. You ask yourself 'Why should I go out and do exercise?', I feel my household chores are already a hand full"
>
> (IDM 005)

**Sociodemographics.** This subcategory describes the impact of social, economic, financial, and cultural paradigms on self-management activities. These narratives demonstrate that the variation in goal-setting behaviour is contingent on the 1) biological age when a diagnosis of T2D is made, 2) level of education, 3) personal religious beliefs, and 4) affordability of sustaining self-care habits.

Most individuals with T2D dismiss the relevance of age in their commitment to stringent disease management. They say that "there is no difference; it all depends on the willingness to follow the doctors' advice". Nevertheless, they agree that some individuals tend to ignore the consequences of their disease when they are young but become increasingly concerned about it later in life.

People with T2D feel that education is not a measure of eagerness to control their disease.

"Look, even if you hold a master's degree, there is no guarantee you will look after your own wellbeing. I think there is no relationship between being a responsible patient and owning a degree. No I don't think so"

(IDM 015)

"A person with little education will look upon the advice of the doctor as the words of the Gospel. They will follow the doctor's advice in a very rigid manner. Just like my mom"

(IDM 012)

Nevertheless, they believe that T2D individuals from a lower educational background are likely to be dependent on their doctors for advice and support. On the other hand, they reckon that more educated people with T2D are well placed to make their own decisions about their disease management.

"These individuals [with greater degree of education] believe they have sufficient knowledge. As a result, they are extremely confident. So, they tend to do things their way. For example, if they consider something is good for their health, they will generally follow through on their decision to go ahead with it"

(IDM 021)

From a cultural standpoint, Muslim patients express having doubts over the 'kosher status' of insulin. Some also feel that taking insulin prevents them from performing religious fasting practices out of the fear of hypoglycemia. From a financial standpoint, many patients cite affordability as a reason for the lack of self blood glucose monitoring at home. They claim that the cost of purchasing glucometer strips can be prohibitive.

"The glucometer strip costs a small fortune you know. 39 ringgit [USD 9.60] for just 25 strips"

(IDM008)

**Food environment.**   The concept of 'food environment' was developed by considering the physical and social landscapes that determine dietary behaviour in people with T2D. In general, this theme explores their opinion about food quality, the role of legislation, and the benefits of a positive food environment at home.

During FGDs, most individuals with T2D appear to have a negative perception about the quality of meals served at food outlets. One individual with T2D blamed "unscrupulous food vendors for adding too much sugar in their cooked meals". Nevertheless, they are compelled to dine out due to the nature of their work and the size of the family unit. They also question the lack of dietary options to choose from that best suit their disease. As a result, most persons with T2D tend not to observe dietary measures (at times seem to condone this "infrequent behaviour") while eating out. To curb this problem, both GPs and people with T2D have suggested "the need for legislative action" to uphold food standards in the country. From a more

personal perspective, T2D individuals also strongly feel that they can achieve recommended dietary goals by regularly consuming meals at home.

> "When we consume meals from food outlets or at the cafeteria while working away from home, we don't really get to choose proper food doctor. So, the reading [blood glucose] will increase and then decrease sometimes"
>
> (IDM002)

> "My wife says she will help prepare a much healthier diet while I am at home. So she ensures we take food which contains more fibre and lesser amounts of sugar. I try to avoid sugary beverages to the best of my abilities. I think it's best not to consume food when I am out of the house because it's much healthier eating at home. We eat more vegetables."
>
> (IDM 014)

## Personal experience

**Adverse effects.**   Individuals with T2D in our study commonly experience minimal side effects with their medications. When it does occur, they cite physical fatigue as the most typical medication adverse effect, followed by dizziness, hypoglycemia, gastrointestinal problems, and loss of productivity. Any noticeable reaction to their medication also leads to self-alteration of the recommended medication dosage.

> "I can still consume this medication, but for me, I feel it makes me feel lethargic and tired"
>
> (IDM 019)

> "The doctor told me to take two tablets right? But I get really feel lethargic and dizzy when I take 2 tablets. So I reduced it to just 1 tablet myself"
>
> (IDM 007)

**Experiential reflections.**   'Experiential reflections' refer to the awareness of a unique set of observations during the course of coping with T2D daily. These observations include the perceived physical reactions associated with 1) fluctuations in disease patterns and 2) medication consumption.

People with T2D feel their bodies respond differently to medications, claiming that sometimes "it takes a longer time for medications to be effective". This subjectivity varies from one individual to another.

> "Based on my own experience, previous drugs didn't really work for me. Only this current drug, I could see changes within the week."
>
> (IDM 004)

> "It may vary from person to person. Some may react differently. For me, I know, 2 tablets is the best for my body."
>
> (IDM 012)

In a similar vein, some T2D individuals with elevated HbA1c levels have a higher degree of sensitivity to the specific physical manifestation (lethargy, frequent micturition,

hypoglycaemia, or body aches) while some of them remain entirely oblivious. These symptoms appear to be more prevalent when they "keep missing medications for a prolonged period of time".

**Uncertainty and wellbeing.**   It is not uncommon for some T2D individuals to exhibit elements of uncertainty and frustration in managing their disease. They invoke feelings of "I feel alright today, but I am not sure about tomorrow" despite all attempts to effectively cope with their disease.

> "Right now, it's under control [diabetes]. But I just feel it can go up [blood glucose level] at any time"
>
> (IDM 002)

> "I know it is important to control my disease. I just feel afraid of the disease sometimes. I am just troubled by the fact that anything [complications] can occur at any time [beyond control]"
>
> (IDM 006)

However, they gain more confidence by observing improved physical wellbeing as a result of consuming either allopathic or complementary alternative medications (CAM).

> "I feel much better right now [after taking medications]. I feel physically healthy and my work performance is much better. I no longer feel lethargic"
>
> (IDM 007)

> "I have not been compliant with my medications, you know, and my blood glucose readings [glucometer] were at 9 or 10. But once I consume blended bitter gourd juice, I realize the readings tend to normalize"
>
> (IDM 10)

**Exemplar and the influence of others.**   The subcategory 'exemplar' represents a person or life event from which a specific behavioural pattern is emulated. In this instance, we discovered that people with T2D use cues derived from peer interactions or life experiences to inform them of their beliefs surrounding self-management practices.

The motivation for a more disciplined approach to T2D control arises from witnessing complications in other people living with T2D. This direct observation triggers a certain level of openness, and they become more receptive to learning from people who manage their diabetes effectively.

> "I have a friend who called me up saying, 'I am doing fine without medications'. A month later, I received a call from that same friend saying, in a distressed voice, that he recently suffered from an episode of stroke. So, medication is something that I must stick to diligently [to prevent complications]"
>
> (IDM 017)

Persons with T2D eventually adopt the position of "role models" to both friends and family members. On the other hand, some acknowledge receiving incorrect advice from several people with T2D or even family members. Although a positive relationship among family

members supports disease management in most cases, some family members unduly influence individuals with T2D to sidestep disease management by avoiding medications.

"I have heard people saying taking too many tablets can kill you at the end of the day"

(IDM 005)

"My family members, especially my sister, she keeps telling me to avoid insulin and get a second opinion instead. When she knows I am going to meet a doctor, she insists that I tell them I am not willing to use insulin"

(IDM 020)

**Relationship with GPs.** The interaction of individuals with T2D with their GPs depends on "which type of doctor" they see on the day of their appointment. They perceive their GPs as agreeable during consultations and consider them as "health advisors". Nevertheless, people with T2D sense unpredictability in the demeanour of GPs, ranging from being "relatively quiet" to being "very approachable". Many of them were disappointed with the attitude of some GPs who appear to be lacking "warmth" or empathy. Small numbers of GPs are seen as noticeably rude, condescending, or even disparaging towards individuals with T2D.

"Some doctors are nice. They go the extra mile. But some doctors just give you the cold shoulder treatment"

(IDM 008)

"There are these doctors who threaten us. They say if we don't take our medication, we might end up dying faster"

(IDM 003)

**Opinion about primary care services.** People with T2D are largely satisfied with the way the primary care clinics operate, stating that the quality of prescribed medications and clinical consultations were remarkable.

"The services provided here is excellent. I have no issues at all"

(IDM 017)

"I will say it [the service provided] is good. Everyone [healthcare providers] really do care and help the patient recover [from diabetes]"

(IDM 013)

"The consultation is good, and they [healthcare providers] do what is really necessary"

(IDM 001)

However, they claim seeing different GPs during each consultation often leaves them confused as they receive conflicting opinions about disease management. Persons with T2D were also disgruntled with the way GPs pay very little attention to their needs because they see too many patients in a day. In the same way, several nurses recalled individuals with T2D complaining that "consultations with the GP were too swift" and doctors "barely had time to address any of my questions".

"Probably because they [GPs] are stressed when they see too many patients. When you see the doctor, he just tells you what your current medical status is like and prescribes you medications. That's about all"

(IDM 002)

"They stare at you, look at your results, and prescribe the same old medications for the next four months. But I don't blame them, they probably have no time"

(IDM 010)

## World view

**Perception of disease.**   People with T2D perceive that diabetes is "becoming a norm" in society. They are acutely aware of the seriousness of the disease in the country. Health care professionals substantiate this notion by stating that persons with T2D often adopt the mentality of "everybody has diabetes nowadays". GPs also feel that individuals with T2D do not have adequate control over their disease. Only a fraction of people with T2D are confident enough to declare they have their disease under control. A certain degree of self-empowerment seems to be lacking to the extent that individuals with T2D place absolute confidence in the doctors' authority.

"Yes. Common. Diabetes is very common in our society"

(IDM 011)

"I don't know whether the medication really helps me or not. I wouldn't know. Because every time I come to the clinic, the doctor looks at me and says 'you have to take this medicine'. So, when you have to take medicine, you just take it. I don't do much research about these things"

(IDM 010)

**Health beliefs.**   Several salient health beliefs about their disease dominate the mindset of individuals with T2D. For example, they feel that T2D can reduce their quality of life as it is closely intertwined with other illnesses such as hypertension and coronary heart disease. On another note, many believe the way sugar is metabolized in the body varies from one individual to another.

There is a prevailing thought that taking too much medication can cause ill health. "Medication can cause kidney damage" is a recurring concern expressed by almost all individuals with T2D. This detail was further corroborated by interviews with GPs as well. People with T2D also feel insulin has many side effects and equate the commencement of insulin to having reached a "critical stage of the disease".

"People consider us as having severely uncontrolled diabetes when we mention that we have started taking insulin"

(IDM 018)

"It's the perception people have you know. Once you start taking insulin, they think that our disease is out of control and that anything can happen to us. It's just a matter of time"

(IDM 004)

A small group of people with T2D believe the disease can be controlled via lifestyle modifications alone. This misconception fuels the need to consume rice sparingly. They also consider the idea of "cooking, sweeping, cleaning and washing" as an equivalent to exercise.

> "I sweep, cook, and do the laundry everyday. I mean when I walk upstairs and down, I feel that is a form of exercise. I sweep the floor and clean the house. And I sweat it out too you know. I don't know it's true or not, but I think that's enough for me."

(IDM 005)

**Knowledge.**   People with T2D commonly associate the cause for T2D with old age, genetic predisposition, deficiency of insulin, and a non-functioning pancreas. They are also cognizant of important complications of T2D, such as kidney damage, coronary artery disease, and neurological disorders.

> "I know diabetes runs in the family. If your dad has it, your mom has it, then 95% of the time you end up getting it too"

(IDM 008)

> "I have this fear that diabetes will lead to kidney problems in the future. It will severely limit my daily activities as I might to undergo hemodialysis when that happens"

(IDM 021)

## Discussion

### Summary of main findings

The results from this study is an extension of a larger qualitative inquiry designed to decipher the intricate web of interaction behind the dynamics of self-management practices in people with T2D [10, 11]. This article demonstrates that self-management behaviour in individuals with T2D consists of a rich interplay between factors associated with their external reality, inner psychological environment, and appropriate behavioural mediators (Fig 1) [10, 11]. More importantly, we have clearly delineated the thoughts and beliefs concerning the external realities that affect the daily lives of people with T2D. We identified three major themes linked to this behaviour: 'intrinsic background status', 'personal experience', and 'world view'. Sub-categories within each theme can be further classified as modifiable (lifestyle, health beliefs, general perception, knowledge, patient-doctor relationship, health services) and non-modifiable (sociodemographic status, the influence of others, food environment) behavioural factors.

The findings in our study have several implications for clinical practice and health policy planning. In the search for real-world solutions to the challenges that arise from their external reality, we analyzed and condensed the complexity of diabetes self-management into actionable health outcomes by using a problem-focused approach. These clinical measures form a part of a logic model that demarcates causal relationships among variables related to their external reality [7]. These measures provide tools for targeted customization aimed at sustained behavioural change that can be used by clinicians during a clinical encounter. On the other hand, public health officials could utilize this information as a source for intervention mapping by setting goals that consist of a balance between modifiable factors (responsive to conventional health promotion strategies) and non-modifiable (possibly requiring a long-term structural reform and legislative process) [7]. Thus, our framework helps 1) summarize this

behavioural intricacies within an operational platform, and 2) establish clear margins amenable to behavioural intervention or preventive measures.

## Strengths and limitations

This study was conducted with participants originating from only one regional primary care institution. However, the richness and the quality of the data were established through the method of triangulation where selected healthcare professionals from adjacent clinics were interviewed to verify the views expressed by the participants. Additionally, we emphasized inclusivity by obtaining participants 1) from a community-based setting and 2) with a mixed sociodemographic background.

## Comparison with literature

The effective management of T2D involves a multifactorial process requiring a focused assessment of different facets of the living environment of people living with T2D [21]. Our study demonstrates that the motivation behind self-management practices in persons with T2D is deeply connected to the understanding of the dynamics of their lifestyle habits. To this end, GPs play an important role in 1) helping patients overcome challenges related to diabetes self-management, and 2) ensuring a mutually beneficial provider-patient interaction that facilitates behavioural change. A person-centered approach should be adopted as a way to encourage GPs to pay more attention to the background status and expectations of the people living with T2D [21].

Our study highlights the paternalistic doctor-patient communication style prevalent in most Southeast Asian countries [22]. Similarly, literature also indicates that health services in these regions are overwhelmed by a high patient load, which could potentially limit constructive consultation time [22]. This situation, in part, gives rise to patients' accounts of grievances with GPs described in the form of 1) having received insufficient advice, 2) inadequate consultation time, and 3) GPs reportedly being poor mannered or inattentive [23]. However, it is equally important to note that a time-constrained clinical environment does not entirely preclude a more participatory role by GPs during any clinical encounter. Instead, what appears to be essential is a high-quality patient-provider communication characterized by non-judgemental acceptance, open and honest interaction [23]. This strategy has consistently shown improvements self-management behaviour in individuals with T2D [23].

The poor conditions and the lack of access to healthier options within the immediate food environment appear to be a constant pre-occupation amongst participants in this study. This study indicates that exposure to "hyperpalatable food" products is associated with food craving tendencies in people with T2D [24]. This phenomenon is evidenced by non-compliance with dietary adherence despite being fully aware of the consequences of their actions. These behavioural patterns suggest a close parallel to evidence from pioneering work from food addiction research. These studies demonstrate a close association between food addiction scores and binge eating severity among persons with T2D [24, 25].

## Implications for future research and clinical implications

The categories described in this study summarizes a realistic profile that shines a light on commonly held doubts and frustrations in people with T2D. These variables serve as a starting point for targeted interventions. Thus, other methods should be employed in conjunction with these findings to increase the likelihood of bringing about behavioural change in persons with T2D. One plausible approach would be the context-specific use of motivational interviewing techniques (promoting the importance of weight control or physical activities) to produce a

more meaningful and empathetic experience for GPs and individuals with T2D [26]. This intervention is likely to improve diabetes-related outcomes when integrated with person-centered self-management education [27]. To a large extent, these positive effects can be further enhanced by addressing negative perceptions and adverse disease experience via group-based self-management education with popular opinion leaders (i.e., T2D individuals with reasonable glycaemic control and expert physicians) using interpretations from real-world parameters depicted in this study [28, 29].

## Conclusion

This study's results are vital to healthcare providers as it underscores common self-management pitfalls faced by people with T2D in their interaction with the healthcare system and their immediate social surroundings. This information can be used to customize the management of their disease by conducting targeted behavioural interventions in accordance with their external realities.

## Supporting information

**S1 File. Topic guides for in-depth interviews and focused group discussions.**
(DOCX)

**S1 Fig. Flow chart of the study design and development of conceptual framework.**
(DOCX)

**S1 Table. Consolidated criteria for reporting qualitative studies (COREQ): 32-item checklist.**
(DOCX)

**S2 Table. Selected quotations from subcategories related to the themes within the external environment.**
(DOCX)

**S3 Table. Themes, sub-themes, codes, quotations within the conceptual model (via Grounded Theory Approach).**
(DOCX)

## Acknowledgments

We would like to thank Monash University of Malaysia for granting us the opportunity and technical support required to carry out this project. The main idea for the conceptual framework in this study was conceived from an epiphany I experienced while on a plane ride from Australia to Malaysia, due in large part to Mr. Rodney Smith's seminal work called 'Awakening: A Paradigm Shift of the Heart'. Finally, we would like to express our heartfelt gratitude to all T2D participants of this study, who were, in all practical sense, the 'true authors' of this manuscript.

## Author Contributions

**Conceptualization:** Yogarabindranath Swarna Nantha.

**Data curation:** Yogarabindranath Swarna Nantha.

**Formal analysis:** Yogarabindranath Swarna Nantha, Azriel Abisheg Paul Chelliah.

**Funding acquisition:** Yogarabindranath Swarna Nantha.

**Investigation:** Yogarabindranath Swarna Nantha, Azriel Abisheg Paul Chelliah, Gan Kim Yen.

**Methodology:** Yogarabindranath Swarna Nantha.

**Project administration:** Yogarabindranath Swarna Nantha, Azriel Abisheg Paul Chelliah.

**Resources:** Yogarabindranath Swarna Nantha.

**Software:** Yogarabindranath Swarna Nantha.

**Supervision:** Yogarabindranath Swarna Nantha, Shamsul Haque, Anuar Zaini Md. Zain.

**Validation:** Azriel Abisheg Paul Chelliah, Gan Kim Yen.

**Visualization:** Yogarabindranath Swarna Nantha.

**Writing – original draft:** Yogarabindranath Swarna Nantha.

**Writing – review & editing:** Yogarabindranath Swarna Nantha.

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
