## [Decision Letter · Decision Letter 0]

13 Oct 2020

PONE-D-20-04931

The External Realities of Individuals with Type 2 Diabetes – Understanding Disease Perspective and Self-Management via Grounded Theory Approach

PLOS ONE

Dear Dr. Swarna Nantha,

Thank you for submitting your manuscript to PLOS ONE. After careful consideration, we feel that it has merit but does not fully meet PLOS ONE’s publication criteria as it currently stands. Therefore, we invite you to submit a revised version of the manuscript that addresses the points raised during the review process.

We look forward to receiving your revised manuscript.

Kind regards,

Janhavi Ajit Vaingankar

Academic Editor

PLOS ONE

Journal Requirements:

2. Please include additional information regarding the interview guide or script used in the study and ensure that you have provided sufficient details that others could replicate the analyses. For instance, if you developed a guide as part of this study and it is not under a copyright more restrictive than CC-BY, please include a copy, in both the original language and English, as Supporting Information.

Reviewers' comments:

Reviewer's Responses to Questions

**Comments to the Author**

1. Is the manuscript technically sound, and do the data support the conclusions?

Reviewer #1: Partly

Reviewer #2: Partly

2. Has the statistical analysis been performed appropriately and rigorously? 

Reviewer #1: N/A

Reviewer #2: N/A

3. Have the authors made all data underlying the findings in their manuscript fully available?

Reviewer #1: No

Reviewer #2: Yes

4. Is the manuscript presented in an intelligible fashion and written in standard English?

Reviewer #1: Yes

Reviewer #2: Yes

5. Review Comments to the Author

Reviewer #1: Thank you for the opportunity to review this manuscript. The study is interesting and reports important findings about the experience of managing type 2 diabetes in adults in Malaysia.

The language throughout the paper is judgmental and needs to be revised using strengths-based, person-centered, and empowering messages. Negative, judgmental language is not only unhelpful to people with diabetes and health care providers, it negates much of what you found in your study. Please see my notes in the attached word document, where I have gone through the entire manuscript, made suggestions, and highlighted concerning examples. Please refer to the guidance published by the American Diabetes Association and Association of Diabetes Care and Education Specialists: Dickinson, et al. (2017). The use of language in diabetes care and education. Diabetes Care, 40, 1790-1799, which includes recommendations for changing specific words/phrases and rationale for doing so.

Only 7 of 18 references represent current literature and a few references are extremely dated.

Your discussion, implications, and conclusions do not sufficiently support the findings nor provide guidance to readers for how to change their practice. Again, your findings are important and I encourage you to consider submitting to another publication once these changes are made and inconsistencies remedied.

Reviewer #2: Dear authors,

thank you very much für the opportunity to read the manuscript „The external realities of Type 2 Diabetes Patients“.

I accepted the review because I was very interested in the title and the method, but unfortunately, more questions remained with me than I received answers. In the following lines, I explain what I mean by this.

I will skip the abstract and the introduction. My comments on this can be read in the overall evaluation.

Material and Methods

„A qualitative grounded theory approach“ ist double formulated. I don’t know a quantitative GT.

Purposive sampling is not the sampling approach for Grounded Theory. This is the theoretical sampling. This deviation should be well justified and should be mentioned in the limitations in my opinion.

Among the participants, it is not quite clear to me why doctors, diabetes pharmacists and pharmacists were questioned. When it comes to self-managment beliefs in T2D patients, the external perspective irritates me. If it needs it, please explain it better. I do not understand it and find it highly problematic. Were the different sources brought together? The title of the paper raises doubts about the usefulness of the expert information.

Why were the same patients questioned in focus groups in addition to the individual interviews? What were the topics? And so it leads straight over to my biggest question or my biggest concern: What did you actually ask in the interviews? You write they are in-depth interviews. That is where a narrative-generating impulse is set. Which one is that? I can't find it in the manuscript. Can you give us your central question. A little later I read "topic-guide". This does not fit to in-depth interviews. That would be semi-structured interviews. Following your advice, I looked at the study protocol and the topic guide. Afterwards I would very much question the term in-depth interview.

Furthermore, you write that you had a theoretical saturation after 10 interviews, but then you explored and enriched this with 14 more interviews. Is it perhaps because of the sampling that so many more interviews had to be conducted?

The description of the analysis does not go beyond the formation of categories. How does the theory emerge?

Results

I don‘ understand how do you observe „thought patterns“.

To Table 3: Experts argue splendidly whether a frequency count of categories is qualitative work. In a Grounded Theory work, I consider it obselete.

Table 3: Why is lifestyle one of the non-modifiable factors?

Chapter lifestyle: „In general, patient frequently miss taking their medikation when they are out of their homes. You cannot introduce such a statement with "in general". What do you base that on, 24 interviews? With this statement, you are overstressing your data.

The presentation of the results seems very descriptive, not very interpretative. So not so much based on Grounded Theory. I miss the merging of the areas; categories are not developed in further details. I cannot reconstruct how Food environment, lifestyle and sociodemographics became the background.

Discussion

I cannot understand the first sentence at all. Where does it come from, I did not read it out at all. I also can't find the content of the second sentence (or the background for it) in the description of the result at all.

The major themes are not comprehensible to me.

Implications: It is not easy for me to deduce possibilities for behavioural change from the categories presented in this paper or to find starting points for targeted interventions.

Overall

The present manuscript describes a small part of the results of the qualitative part of a mixed-methods study. The knowledge gained with the results presented here seems to me to be limited and I do not have the impression that it would help me in science or in practice. The result of the entire qualitative part could be interesting, but not this excerpt alone.

In addition, good mixed-method research is characterised by an integration of qualitative and quantitative parts on as many levels as possible (research question, data collection, analysis). Thus, separate presentation naturally remains below the actual possibilities.

The interview guide, with its strong focus on medication, does not seem to me to be so well aligned with the cognitive interest of the overall project. How the answers to these questions are to be used to develop a theoretical model that explains disease management in diabetics is questionable.

With these estimations, I come back to the introduction. Strictly speaking, it already seems to fit to manuscript. But I as a reader remain somewhat confused with my question: what is the overall question of the project, what is the question of the manuscript, how can the result presented in the manuscript contribute to the overall question.Mediators, categories etc. are mentioned in the paper from time to time, which become relevant in the overall project, but are not connected to the external realities for the time being. This is somewhat confusing.

I do not think that extracting this manuscript is a good idea or that it really needs to be presented more clearly. There are simply too many questions for me as a reader.

6. PLOS authors have the option to publish the peer review history of their article (what does this mean?). If published, this will include your full peer review and any attached files.

Reviewer #1: No

Reviewer #2: No

---

## [Author Response · Author response to Decision Letter 0]

23 Oct 2020

We have attached a detailed response to the comments of the reviewer in a separate file labeled "Response to Reviewer" in the PloS One submission portal

---

## [Decision Letter · Decision Letter 1]

8 Dec 2020

PONE-D-20-04931R1

The External Realities of People with Type 2 Diabetes – Understanding Disease Perspective and Self-Management Behaviour via Grounded Theory Approach

PLOS ONE

Dear Dr. Swarna Nantha,

Thank you for submitting your manuscript to PLOS ONE. After careful consideration, we feel that it has merit but does not fully meet PLOS ONE’s publication criteria as it currently stands. Therefore, we invite you to submit a revised version of the manuscript that addresses the points raised during the review process.

I urge the authors to address the comments from the second reviewer regarding the study design, data collection and results carefully in your revision.

We look forward to receiving your revised manuscript.

Kind regards,

Janhavi Ajit Vaingankar

Academic Editor

PLOS ONE

Reviewers' comments:

Reviewer's Responses to Questions

**Comments to the Author**

1. If the authors have adequately addressed your comments raised in a previous round of review and you feel that this manuscript is now acceptable for publication, you may indicate that here to bypass the “Comments to the Author” section, enter your conflict of interest statement in the “Confidential to Editor” section, and submit your "Accept" recommendation.

Reviewer #1: All comments have been addressed

Reviewer #2: (No Response)

2. Is the manuscript technically sound, and do the data support the conclusions?

Reviewer #1: Yes

Reviewer #2: No

3. Has the statistical analysis been performed appropriately and rigorously? 

Reviewer #1: N/A

Reviewer #2: N/A

4. Have the authors made all data underlying the findings in their manuscript fully available?

Reviewer #1: No

Reviewer #2: No

5. Is the manuscript presented in an intelligible fashion and written in standard English?

Reviewer #1: Yes

Reviewer #2: Yes

6. Review Comments to the Author

Reviewer #1: Thank you for addressing reviewer feedback and changing the language in your manuscript. I hope you will continue to use person-centered, strengths-based, and empowering language going forward in speaking and writing.

Reviewer #2: Many thanks for the careful revision and the extensive explanations in the revision letter. A lot was explained to me, but unfortunately not everything convinced me.

I take a very critical view of the topic guide for the interviews with their strong focus on medication.

Regarding my problem with the inclusion of health care professionals, I was referred to a chapter in the manuscript. However, it remains open for me what external parties can contribute to the questions of the study and what questions were asked to the experts in their individual interviews. None of this clarified the revision for me. Somewhere in the "strenghs and limitations" I am reading that the statements of people with diabetes were submitted to the experts for verification. Why should this happen when it comes to the "realities" of people with diabetes.

In the results section I miss the framing. Incredibly many small individual dimensions are opened and quotes are added. But how they (can be) brought together in the larger dimension is hardly comprehensible for me.

I maintain that it seems unfavorable to me to publish this paper, which is separated from the overall results. In my opinion, this leaves too much open for the reader and the practical benefit seems doubtful to me.

7. PLOS authors have the option to publish the peer review history of their article (what does this mean?). If published, this will include your full peer review and any attached files.

Reviewer #1: No

Reviewer #2: No

---

## [Author Response · Author response to Decision Letter 1]

17 Dec 2020

Our comments are included in the "Response to reviewers" file uploaded to the web portal.

---

## [Decision Letter · Decision Letter 2]

22 Dec 2020

The External Realities of People with Type 2 Diabetes – Understanding Disease Perspective and Self-Management Behaviour via Grounded Theory Approach

PONE-D-20-04931R2

Dear Dr. Swarna Nantha,

We’re pleased to inform you that your manuscript has been judged scientifically suitable for publication and will be formally accepted for publication once it meets all outstanding technical requirements.

Kind regards,

Janhavi Ajit Vaingankar

Academic Editor

PLOS ONE

Additional Editor Comments (optional):

Reviewers' comments:

Reviewer's Responses to Questions

**Comments to the Author**

1. If the authors have adequately addressed your comments raised in a previous round of review and you feel that this manuscript is now acceptable for publication, you may indicate that here to bypass the “Comments to the Author” section, enter your conflict of interest statement in the “Confidential to Editor” section, and submit your "Accept" recommendation.

Reviewer #1: All comments have been addressed

2. Is the manuscript technically sound, and do the data support the conclusions?

Reviewer #1: Yes

3. Has the statistical analysis been performed appropriately and rigorously? 

Reviewer #1: I Don't Know

4. Have the authors made all data underlying the findings in their manuscript fully available?

Reviewer #1: Yes

5. Is the manuscript presented in an intelligible fashion and written in standard English?

Reviewer #1: Yes

6. Review Comments to the Author

Reviewer #1: Thank you for addressing the reviewer's feedback. I have nothing further to add at this time. Thank you.

7. PLOS authors have the option to publish the peer review history of their article (what does this mean?). If published, this will include your full peer review and any attached files.

Reviewer #1: No

---

## [Editor Report · Acceptance letter]

6 Jan 2021

PONE-D-20-04931R2 

The External Realities of People with Type 2  Diabetes – Understanding Disease Perspective and Self-Management Behaviour via Grounded Theory Approach 

Dear Dr. Swarna Nantha:

I'm pleased to inform you that your manuscript has been deemed suitable for publication in PLOS ONE. Congratulations! Your manuscript is now with our production department. 

Kind regards, 

on behalf of

Ms Janhavi Ajit Vaingankar 

Academic Editor

PLOS ONE